# Swin-LiteMedSAM: A Lightweight Box-Based Segment Anything Model for Large-Scale Medical Image Datasets

Ruochen Gao[†][⋆][0000−0002−9411−3369], Donghang Lyu[†][⋆][0009−0008−1446−9930], and Marius Staring[†][0000−0003−4045−9732]

[†]Division of Image Processing, Department of Radiology, Leiden University Medical Center, Leiden, the Netherlands
{r.gao, d.lyu}@lumc.nl

**Abstract.** Medical imaging is essential for the diagnosis and treatment of diseases, with medical image segmentation as a subtask receiving high attention. However, automatic medical image segmentation models are typically task-specific and struggle to handle multiple scenarios, such as different imaging modalities and regions of interest. With the introduction of the Segment Anything Model (SAM), training a universal model for various clinical scenarios has become feasible. Recently, several Medical SAM (MedSAM) methods have been proposed, but these models often rely on heavy image encoders to achieve high performance, which may not be practical for real-world applications due to their high computational demands and slow inference speed. To address this issue, a lightweight version of the MedSAM (LiteMedSAM) can provide a viable solution, achieving high performance while requiring fewer resources and less time. In this work, we introduce Swin-LiteMedSAM, a new variant of LiteMedSAM. This model integrates the tiny Swin Transformer as the image encoder, incorporates multiple types of prompts, including box-based points and scribble generated from a given bounding box, and establishes skip connections between the image encoder and the mask decoder. In the *Segment Anything in Medical Images on Laptop* challenge (CVPR 2024), our approach strikes a good balance between segmentation performance and speed, demonstrating significantly improved overall results across multiple modalities compared to the LiteMedSAM baseline provided by the challenge organizers. Our proposed model achieved a DSC score of **0.8678** and an NSD score of **0.8844** on the validation set. On the final test set, it attained a DSC score of **0.8193** and an NSD score of **0.8461**, securing fourth place in the challenge. The code and trained model are available at https://github.com/RuochenGao/Swin_LiteMedSAM.

**Keywords:** LiteMedSAM · Swin Transformer · Multiple Prompts.

---

[⋆] These authors contributed equally to this work.

## 1   Introduction

Medical imaging diagnosis is fundamental for evaluating diseases, and medical image segmentation, which involves the extraction of specific structures such as tumors and organs from medical images, consistently receives significant attention. Deep learning methods have demonstrated effectiveness in this field, leading to the development of numerous models tailored for specific scenarios. However, each scenario typically requires training a dedicated segmentation model, demanding substantial effort. In recent years, inspired by the rapid development of large language models (LLMs) in the natural language processing (NLP) field, researchers have begun exploring the application of large models in computer vision. Segment Anything Model (SAM) [5] is one such innovation, aiming to unify the segmentation task for general images by training with a huge amount of data. while SAM holds potential, the distinct features of medical images can hinder its performance in medical image segmentation. Therefore, recent works [8,10] focus on adapting the SAM model for medical applications by re-training with a large volume of medical images. Despite achieving high performance in various medical image segmentation tasks, SAM models' large parameter volume and the high spatial resolution of medical images require substantial computational resources and processing time. This poses challenges for practical deployment of SAM models in real-world applications, or even for non-industry academic groups conducting research on them. Consequently, lite SAM models are gaining more attention as a solution to this problem.

The original SAM model is composed of three main components: an image encoder, a prompt encoder, and a mask decoder. Among these, the image encoder is the primary factor contributing to high computational and memory costs due to the usage of ViT-H [3]. To mitigate resource consumption and accelerate processing, various studies have aimed to make the image encoder more lightweight. For instance, FastSAM [15] introduces a CNN-based framework, while Mobile-SAM [13] tackles this issue by distilling knowledge from the ViT-H image encoder into a tiny ViT-based encoder. Additionally, EfficientSAM [11] employs the Masked Autoencoders (MAE) [4] framework to efficiently transfer knowledge from a large image encoder to a small one, resulting in a more resource-efficient design with better performance. EfficientViT-SAM [14] further enhances this approach by incorporating EfficientViT [1] with fused MBConv blocks [9] to create a lightweight image encoder. Recently, the challenge *Segment Anything in Medical Images on Laptop*[1], hosted at CVPR 2024, sought universal promptable medical image segmentation models deployable on laptops or edge devices without GPU reliance. The organizers developed LiteMedSAM[2] as a baseline, using the distillation strategy described in [13]. Although LiteMedSAM focuses on optimizing the image encoder to reduce resource usage, segmentation performance is compromised. Therefore, our goal is to enhance performance without highly sacrificing efficiency. To achieve this, we use a lightweight Swin Trans-

---

[1] https://www.codabench.org/competitions/1847/
[2] https://github.com/bowang-lab/MedSAM/tree/LiteMedSAM

former as image encoder and also introduce two additional prompts, box-based points and box-based scribble, except the original box prompt. To this end, we introduce our model, Swin-LiteMedSAM. The key contributions of our model are as follows:

– Instead of transferring knowledge to a tiny ViT, we employ a tiny Swin Transformer [6] as the image encoder. The Swin Transformer is designed to handle large images more efficiently, both in terms of computation and memory usage compared to ViT. Moreover, skip connections are established between the image encoder and mask decoder to enhance feature integration.
– We introduce additional types of prompts beyond boxes, including box-based points and box-based scribble. These prompts are automatically generated from the given bounding box and effectively improve model performance without significantly increasing resource costs.
– Overall, Swin-LiteMedSAM achieves substantial improvements in performance over LiteMedSAM while maintaining high inference speed.

## 2  Method

### 2.1  Data preprocessing

To accelerate the model's training and inference stages and reduce memory consumption, we resize the input image to $256 \times 256$. This is achieved by first resizing the images while maintaining their original aspect ratio based on the longest side, and then do zero padding to reach the final size of $256 \times 256$. For data normalization, we use the method described in [8]. Please refer to [8] for more details.

Note that gray-scale images such as CT, MR, US, and PET typically have only one channel, whereas RGB images from modalities like endoscopy, dermoscopy, and fundus imaging usually have three channels. To maintain consistency during model training, we replicated the channel dimension for gray-scale images, converting them from one channel to three channels.

### 2.2  Proposed method

Our model's structure is shown in Fig. 1. It mainly comprises three components: an image encoder, a prompt decoder, and a mask decoder. The function of these three components are detailed below.

The image encoder architecture is inspired by the original tiny ViT design of LiteMedSAM. The input first passes through two convolutional layers, which capture low-level spatial features and adjust the number of channels to 64. Following this, the encoder consists of four stages, with their depths arranged according to the tiny ViT configuration as (2, 2, 6, 2). The structure of the Swin block used in our encoder is illustrated in Fig. 2. We have slightly modified the standard Swin block by adding a convolutional block with batch normalization between the windowed multi-head self-attention (W-MSA) module and

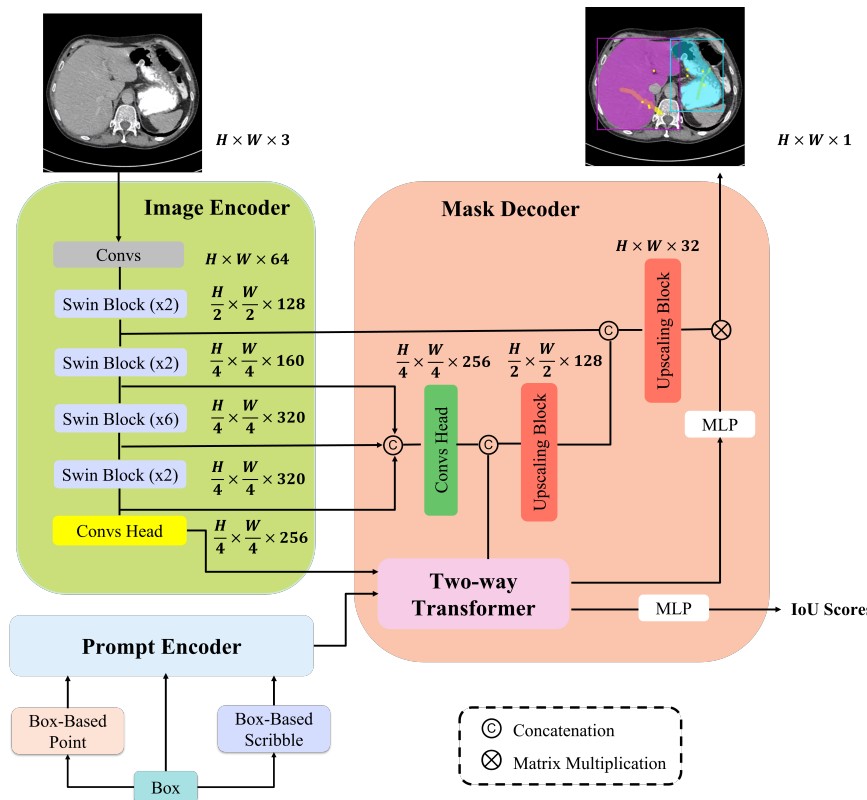

**Fig. 1.** Overview of the Swin-LiteMedSAM architecture.

the Multi-Layer Perceptron (MLP). This modification enables our encoder to effectively capturing both global and local features. Furthermore, the number of channels and spatial resolution across four stages remain consistent with the original design. Finally, a head branch consisting of several convolutional layers and layer normalization adjusts the channel number to 256.

In the prompt encoder, we introduce two additional types of prompts: box-based points and a box-based scribble, alongside the original box prompt. The box-based points and the box are combined to form a sparse embedding, while the box-based scribble is used for dense embedding. For the box-based prompt, drawing from insights provided by [10] and [2], which demonstrate the effectiveness of using multiple points over a single point, we opt to utilize four points in our prompt encoder. To achieve this, we divide the bounding box area into four equivalent sub-parts based on the central point. We then randomly generate one point in the non-zero area of each sub-part, resulting in four points distributed inside the box. If a sub-part contains only zeros, we select the central point. This approach ensures a relatively sparse distribution of points that covers more area.

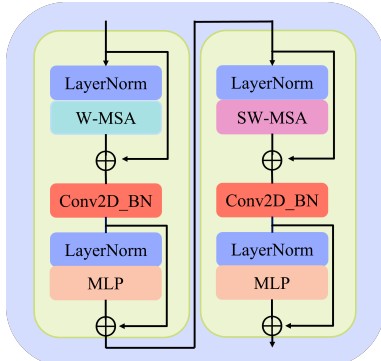

**Fig. 2.** The overall structure of the Swin Transformer block.

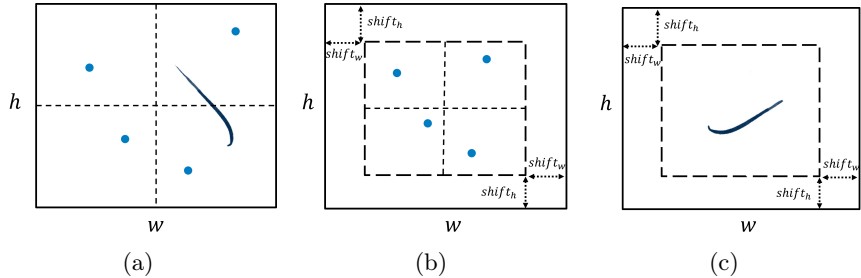

**Fig. 3.** (a) Box-based points and scribble generation strategies during the training stage. (b) Box-based points generation strategy during the inference stage. (c) Box-based scribble generation strategy during inference stage.

Furthermore, a box-based scribble is randomly generated within the box using the algorithm in [16]. All pixels in the scribble are set to 1 and placed into the corresponding part of an all-zero matrix with a shape of (256, 256) to create a mask for the dense embedding. Similarly, if all pixels in the box are zeros, the scribble is set to an all-zero matrix of shape (256, 256) to ensure the prompt encoder focuses on the sparse prompt embedding part, as illustrated in Fig. 3(a).

Then in the mask decoder, we follow the SAM original design by using a two-way transformer to process embeddings from the prompt encoder and image encoder. Moreover, we build skip connections between the image encoder and mask decoder, concatenating outputs from the last three stages and fusing them with several convolutional layers. This output is then combined with the two-way transformer's output and passed through an upscaling block to double the image resolution. Similarly, the upscaled output is concatenated with the first stage's output from the image encoder, and the resulting output is further upsampled to return back to the original spatial resolution.

For the loss function, it consists of a mask prediction loss $L_{mask}$ and a IoU score prediction loss $L_{iou}$:

$$L = L_{mask} + L_{iou}, \tag{1}$$

where $L_{mask}$ is the summation of the Dice loss and binary cross-entropy by comparing the predicted mask with the ground truth mask, while $L_{iou}$ is the MSE loss between the predicted and actual IoU scores.

### 2.3   Inference strategy for box-based points and box-based scribble

The strategy for generating box-based points and box-based scribble has some difference between the training and inference stages. During the training stage, the range for generating four points is within the entire bounding box, which aims to expose the model to diverse cases and helps improve its generalization capabilities. However, randomly generating points within the whole box might not be ideal during the inference stage, as objects are typically located near the central part of the box. Random generation can easily place some points near the boundary, which is less effective and even negatively impact performance. Furthermore, for the situation of a single point prompt, the central point of the box is always the first choice [2]. Likewise, the two corner points of the box already provide some external and surrounding information of objects. Therefore, points should be better distributed in the relatively central part of the box. For a given bounding box, represented by its upper left point $(x_{min}, y_{min})$ and bottom right point $(x_{max}, y_{max})$, we introduce two variables, $\text{shift}_h$ and $\text{shift}_w$, to adjust the coordinates along height and width directions so that four points do not occur in the peripheral area, as shown in Fig. 3(b). This adjustment is denoted as follows:

$$x'_{min} = x_{min} + \text{shift}_w,$$
$$y'_{min} = y_{min} + \text{shift}_h,$$
$$x'_{max} = x_{max} - \text{shift}_w,$$
$$y'_{max} = y_{max} - \text{shift}_h.$$

Here, the new upper left point $(x'_{min}, y'_{min})$ and bottom right point $(x'_{max}, y'_{max})$ form a new box and randomly generate four points within it. The range of $\text{shift}_w$ is $(0, \frac{1}{2}w)$, and the range of $\text{shift}_h$ is $(0, \frac{1}{2}h)$, where $w$ and $h$ are the width and height of the image, respectively. In this study, we adjusted the range of $\text{shift}_w$ to $(\frac{1}{5}w, \frac{2}{5}w)$ and the range of $\text{shift}_h$ to $(\frac{1}{5}h, \frac{2}{5}h)$ to ensure that the distribution of points is closer to the center. Additionally, $\text{shift}_w$ and $\text{shift}_h$ are randomly adjusted within their ranges for each sample to achieve better overall performance. We also follow the same points distribution strategy as in the training stage to ensure that the four points are positioned in the four quadrants of the image.

Then Fig. 3(c) illustrates the strategy of generating a scribble in the inference stage. Considering the empirical distribution of points, we believe that placing the scribble closer to the edges is more effective than points for capturing contour information. Therefore, we adjusted the range of $\text{shift}_w$ to $(\frac{1}{8}w, \frac{1}{6}w)$ and the

**Table 1.** Evaluation Platform environment settings.

| | |
|---|---|
| System | Ubuntu 20.04 Desktop |
| CPU | Intel Xeon(R) W-2133 @3.60GHz |
| RAM | 8GB |
| Docker version | 20.10.13 |

**Table 2.** Training environmentn settings.

| | |
|---|---|
| System | Red Hat 9 |
| CPU | AMD EPYC 7513 @2.60GHz |
| RAM | 256GB |
| GPU (number and type) | One NVIDIA A100 40G |
| CUDA version | 12.4 |
| Programming language | Python 3.10 |
| Deep learning framework | PyTorch 2.2.2 |
| Specific dependencies | Monai, Einops, Timm and Transformers |

range of $\text{shift}_h$ to $(\frac{1}{8}h, \frac{1}{6}h)$ to expand the area for generating a scribble. Note that we generate the scribble in non-zero areas, based on the prior knowledge that people typically avoid drawing scribble in regions with zero pixel values.

## 3  Experiments

### 3.1  Dataset and evaluation metrics

**Training and validation dataset** We only use the provided challenge dataset, without additional public datasets. This dataset includes 11 modalities: CT, MRI, PET, X-ray, ultrasound, mammography, OCT, endoscopy, fundus, dermoscopy, and microscopy, totaling more than one million 2D image-mask pairs.
**Testing dataset** The testing set in this challenge is hidden, with all testing images newly collected from 20+ different institutions worldwide.
**Evaluation metrics** The evaluation metrics are the Dice Similarity Coefficient (DSC) and Normalized Surface Dice (NSD) for accuracy, and Docker container running time for efficiency. These metrics together determine the ranking. Note that only mean results are available. The evaluation platform environment is presented in Table 1.

### 3.2  Implementation details

**Training environment settings** The training environments are presented in Table 2.
**Training protocols** Our training strategy consists of two stages. In the first stage, we utilize knowledge distillation to transfer information from the large ViT-B image encoder to the tiny Swin Transformer as our image encoder. To note, we pre-saved the output image embeddings from the ViT-B encoder to

**Table 3.** Training protocols of the first stage and the second stage.

|  | The first stage | The second stage |
|---|---|---|
| Pre-trained Model | MedSAM ViT-B | Tiny Swin Transformer |
| Batch size | 64 | 16 |
| Patch size | 256×256×3 | 256×256×3 |
| Total epochs | 10 | 25 |
| Optimizer | AdamW | AdamW |
| Initial learning rate (lr) | 2e-4 | 2e-4 |
| Lr decay schedule | ReducedLROnPlateau | ReducedLROnPlateau |
| Training time | 60.8 hours | 46 hours |
| Loss function | L1 Loss | MSE Loss+Dice Loss+BCE Loss |
| Number of model parameters | 10.51M | 36.77M |
| Number of flops | 47.70G | 55.20G |

speed up the distillation process. In the second stage, we take the pre-trained image encoder from the first stage and proceed to train the entire model. The training details of these two stages are listed in Table 3.

**Data sampling strategy** During the training, we randomly sample image cases from the dataset. If the case is 3D, such as a CT, MR, or PET scan, we randomly sample a slice from the 3D image. If the case is 2D, such as an X-ray or microscopy image, we use the image directly. This strategy significantly reduces training time and ensures a more balanced distribution of training samples across different modalities.

**Data augmentation** We apply vertical and horizontal flips to the image, each with a 50% probability.

**Inference environment settings** During the inference stage, the running environment differs from the training stage. A docker container is built, starting with a 'python:3.10-slim' image and installing the CPU version of PyTorch 2.2.2. All other aspects still remain same with the training stage.

## 4    Results and discussion

### 4.1    Quantitative results on validation set

Table 4 shows that Swin-LiteMedSAM achieves higher average DSC (86.70%) and NSD (88.55%) scores compared to LiteMedSAM, which recorded 83.81% for DSC and 83.26% for NSD. In general, Swin-LiteMedSAM achieved a more balanced and comprehensive performance across the nine modalities compared to LiteMedSAM. It showed significant improvement in PET and Microscopy while maintaining strong performance in most modalities. However, the model experienced a noticeable drop in DSC and NSD scores for the US modality.

Then Table 5 further highlights the importance of each component in our proposed method, particularly the inclusion of skip connections, as well as both

**Table 4.** Comparison between LiteMedSAM and our proposed Swin-LiteMedSAM.

| Target | LiteMedSAM | | Swin-LiteMedSAM | |
|---|---|---|---|---|
| | DSC (%) | NSD (%) | DSC (%) | NSD (%) |
| CT | **92.26** | **94.90** | 91.46 | 94.70 |
| MR | **89.63** | **93.37** | 87.12 | 91.19 |
| PET | 51.58 | 25.17 | **69.43** | **56.99** |
| US | **94.77** | **96.81** | 85.57 | 90.63 |
| X-ray | 81.05 | 85.35 | **83.98** | **88.88** |
| Dermoscopy | 92.47 | 93.85 | **94.20** | **95.65** |
| Endoscopy | **96.04** | **98.11** | 95.29 | 97.63 |
| Fundus | 94.81 | 96.41 | **95.83** | **97.39** |
| Microscopy | 61.63 | 65.38 | **77.45** | **83.91** |
| Average | 83.81 | 83.26 | **86.70** | **88.55** |

**Table 5.** Ablation study of the proposed method. The check mark shows including the module in the method. Here, Swin-T indicates tiny Swin Transformer.

| Swin-T | Skip connection | Box-based points | Box-based scribble | DSC (%) | NSD (%) |
|---|---|---|---|---|---|
| ✓ | | | | 85.79 | 86.75 |
| ✓ | ✓ | | | 86.48 | 87.74 |
| ✓ | ✓ | ✓ | | 86.22 | 87.79 |
| ✓ | ✓ | ✓ | ✓ | **86.70** | **88.55** |

box-based points and scribble, in achieving superior segmentation performance. Here, the introduction of two additional box-based prompts provide limited improvement. This could be due to two factors. First, some prompts may have been placed in sub-optimal positions due to the random way, negatively impacting overall performance. Second, inadequate training can be a contributing factor. Although the data sampling strategy helped balance the distribution of modalities and accelerated the training process, it significantly reduced the number of training samples. This reduction can hinder the effective training of the prompts, which require a high volume of diverse cases to perform optimally.

## 4.2   Quantitative results on testing set

As shown in Table 6, our proposed method significantly outperforms LiteMedSAM across most imaging modalities in terms of DSC and NSD, while also reducing runtime of all the modalities. Specifically, for CT images, our method achieved an absolute DSC improvement of 17.15%, corresponding to a relative improvement of 30.76%, and an NSD increase of 18.51%, corresponding to a relative improvement of 31.75%, compared to LiteMedSAM, also with a faster runtime. For PET and X-ray modalities, our method demonstrated competitive DSC and NSD results. In PET, it achieved a marginal NSD improvement while maintaining similar DSC performance, and significantly reduced runtime. For X-ray, despite a slightly lower DSC compared to LiteMedSAM, the difference is minimal, demonstrating a still competitive result.

**Table 6.** Quantitative evaluation results for final testing set

| Target | LiteMedSAM[3] | | | Proposed Swin-LiteMedSAM | | |
|---|---|---|---|---|---|---|
| | DSC (%) | NSD (%) | Runtime (s) | DSC (%) | NSD (%) | Runtime (s) |
| CT | 55.75 | 58.48 | 32.68 | **72.90** | **76.99** | **25.14** |
| MR | 64.80 | 62.75 | 15.91 | **68.61** | **70.13** | **13.44** |
| PET | **76.94** | 66.98 | 12.99 | 76.50 | **67.63** | **10.52** |
| US | 85.24 | 89.73 | 8.27 | **88.01** | **92.43** | **7.58** |
| X-ray | **85.51** | **94.40** | 8.79 | 84.58 | 94.32 | **6.89** |
| Endoscopy | 94.41 | 96.95 | 13.85 | **94.58** | **97.17** | **11.36** |
| Fundus | **87.47** | **89.58** | 11.72 | 80.71 | 82.93 | **9.85** |
| Microscopy | 84.36 | 86.15 | 11.85 | **87.08** | **88.94** | **10.48** |
| OCT | 73.31 | 80.20 | 8.39 | **84.39** | **90.97** | **6.87** |
| Average | 78.64 | 80.58 | 13.99 | **81.93** | **84.61** | **11.01** |

Furthermore, we observed significant instability in the original LiteMedSAM. Taking CT modality as an example, LiteMedSAM performed exceptionally well on the validation set, surpassing Swin-LiteMedSAM. However, when evaluated on the testing set, performance of CT experienced a significant performance drop, with DSC falling from 92.26% to 55.75% and NSD dropping from 94.90% to 58.48%. Although Swin-LiteMedSAM encounters a similar issue with the CT modality, the performance drop is much less severe. Furthermore, this issue is observed in other modalities as well, further approving that the Swin-LiteMedSAM model offers better stability and generalization, which are essential for the real world applications.

### 4.3   Qualitative results on external public dataset

Since the ground truth for the challenge validation and testing set is not available, we select SegRap2023 [7], a public head and neck CT dataset containing annotations for multiple organs, to verify the model's performance.

As depicted in Fig. 4, we showcase three representative examples from Seg-Rap2023 to visually check our model's performance. In the first case, our model demonstrates strong performance in brain segmentation. This is primarily attributed to the brain's large size and distinct contrast with surrounding tissues. Moving to the second case, we observed that our model maintains good performance even with smaller targets such as the spinal cord, esophagus, and trachea. However, in the third case, our model's performance falls short compared to the ground truth. The main issue arises from the ambiguous semantics in medical images. For instance, when aiming to segment the oral cavity, our method only identifies the teeth. This discrepancy stems from the fact that the box prompt for oral cavity can also be interpreted as segmenting teeth. It is hard to provide a more precise prompt in this case to specify the intended target for segmentation.

---

[3] The model weights and results are released by the challenge organizer.

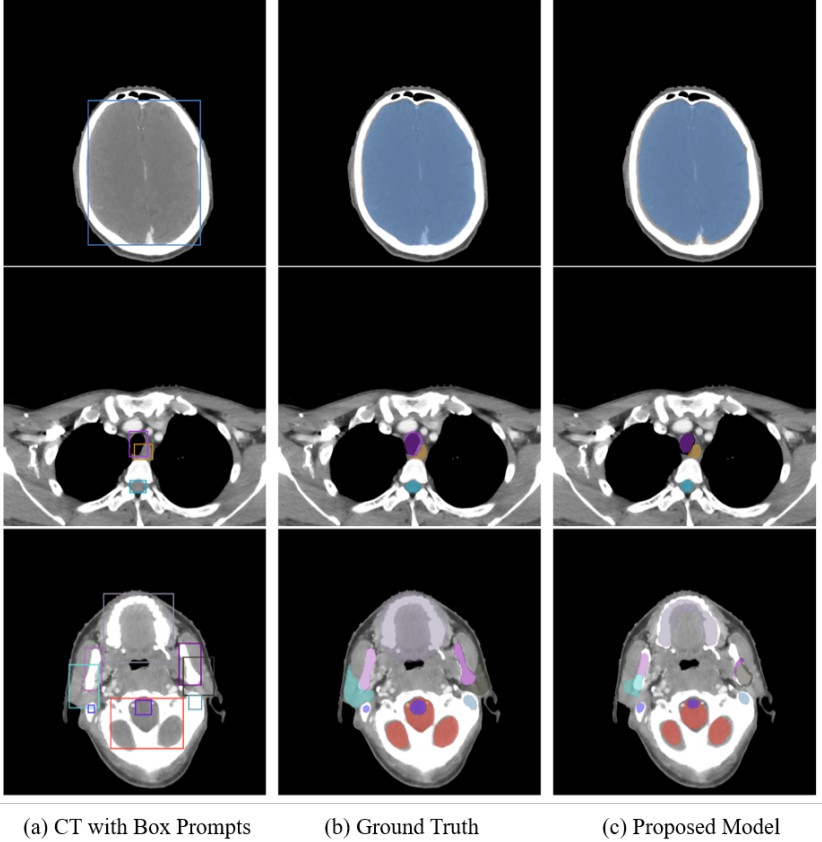

(a) CT with Box Prompts          (b) Ground Truth          (c) Proposed Model

**Fig. 4.** Visual comparison between ground truth and our proposed method, with each row representing one case from SegRap2023. (a), (b), and (c) represent the original image with box prompts, ground truth, and the prediction results of our proposed model, respectively.

### 4.4    Limitation and future work

One main limitation of our method for this challenge is that our model is using 2D images for training and validation, whereas medical imaging data, such as CT, MRI, and PET, are typically in 3D format. Currently, we process these 3D images by making predictions on individual 2D slices, which does not fully utilize the 3D anatomical information and might hinder the performance improvement. The key issue is that the prompts input to the model are generally based on 2D information, such as bounding boxes and points. In the future, we aim to explore how to provide effective prompt information in 3D and adapt the model to handle 3D images directly.

Additionally, we applied certain manual rules to control the distribution of box-based points and the scribble, which is impossible to find the optimal setting

and can easily do harm to the overall performance if not set properly. Furthermore, due to variations in medical modalities and the shapes of segmentation targets, the distribution of points and scribble should be adjusted accordingly. Therefore, developing a learning-based method for generating box-based points and scribble would be highly beneficial and could further enhance the model's performance.

## 5      Conclusion

In this paper, we introduce Swin-LiteMedSAM, a lightweight box-based segment anything model. Our model utilizes the tiny Swin Transformer as image encoder, enabling it to extracts high-level features more effectively. Additionally, the introduction of box-based points and box-based scribble provide more spatial cues, which improve segmentation accuracy without substantially increasing computational costs or demanding extensive manual annotation. Overall, our approach achieves stronger and more stable performance across different medical imaging modalities while maintaining fast inference speed, outperforming the LiteMedSAM model.

**Acknowledgements**  This study utilized computing resources from the Academic Leiden Interdisciplinary Cluster Environment (ALICE) provided by Leiden University. We thank all the data owners for making the medical images publicly available and CodaLab [12] for hosting the challenge platform. This work was supported by the China Scholarship Council (No. 202207720085) and the project ROBUST: Trustworthy AI-based Systems for Sustainable Growth with project number KICH3.LTP.20.006, which is (partly) financed by the Dutch Research Council (NWO), Philips Research, and the Dutch Ministry of Economic Affairs and Climate Policy (EZK) under the program LTP KIC 2020-2023.

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
