# OpenReview forum: "Swin-LiteMedSAM: A Lightweight Box-Based Segment Anything Model for Large-Scale Medical Image Datasets"
_thecvf.com/CVPR/2024/Workshop/MedSAMonLaptop — CVPR24 MedSAMonLaptop_

### Official Review · Reviewer_7ADu · 2024-06-14
**Effective Use of Swin Transformer with High Accuracy and Efficiency**

**Rating:** 10
**Confidence:** 5

**Review:**

I am impressed by the use of the Swin transformer to accelerate inference through the application of appropriate training protocols, achieving high accuracy. However, I believe there is a need for more detailed information about the dataset. Providing a comprehensive overview of the sources of the various datasets used in the challenge would greatly enhance the clarity and replicability of the research.

---

### Official Review · Reviewer_R5KR · 2024-06-22
**Accepted subject minor revision**

**Rating:** 8
**Confidence:** 4

**Review:**

1. The author's ORCID is missing.
2. The equation on page 6 is not numbered.
3. The complete code link is not provided.

---

### Official Review · Reviewer_M8Pe · 2024-06-24
**great paper but need to add more details**

**Rating:** 7
**Confidence:** 5

**Review:**

- Fig. 2. Please add more details on the network modules to caption, e.g., W-MSA, SW-MSA
- The authors mentioned that "we opt for a tiny Swin Transformer [6] as our image encoder, which is designed to handle large images
more efficiently compared to ViT in terms of both computation and memory consumption.". However, in Table 8, the proposed method is slower than the ViT encoder for some modalities. Please explain the potential reason.
- It would be great to try OpenVINO to speed up the model inference.

---

### Official Review · Reviewer_m44M · 2024-06-24
**Swin Transformer and diverse prompt enhances the efficiency and performance of MedSAM.**

**Rating:** 9
**Confidence:** 4

**Review:**

The paper proposes using the Swin Transformer for efficiency and the addition of point and scribble prompts to assist the model training and inference.
1. The link to the code is missing.
2. Adding how the weights of the prompt encoder and the mask decoder in the second stage of the model training would be helpful.

---

### Decision · Program_Chairs · 2024-10-01

Accept